# MVTokenFlow: High-quality 4D Content Generation using Multiview Token Flow

**Hanzhuo Huang**[1]*, **Yuan Liu**[3]*, **Ge Zheng**[1], **Jiepeng Wang**[4], **Zhiyang Dou**[4], **Sibei Yang**[2]†

[1]ShanghaiTech University, [2]Sun Yat-sen University,
[3]The Hong Kong University of Science and Technology, [4]The University of Hong Kong

## Abstract

In this paper, we present MVTokenFlow for high-quality 4D content creation from monocular videos. Recent advancements in generative models such as video diffusion models and multiview diffusion models enable us to create videos or 3D models. However, extending these generative models for dynamic 4D content creation is still a challenging task that requires the generated content to be consistent spatially and temporally. To address this challenge, MVTokenFlow utilizes the multiview diffusion model to generate multiview images on different timesteps, which attains spatial consistency across different viewpoints and allows us to reconstruct a reasonable coarse 4D field. Then, MVTokenFlow further regenerates all the multiview images using the rendered 2D flows as guidance. The 2D flows effectively associate pixels from different timesteps and improve the temporal consistency by reusing tokens in the regeneration process. Finally, the regenerated images are spatiotemporally consistent and utilized to refine the coarse 4D field to get a high-quality 4D field. Experiments demonstrate the effectiveness of our design and show significantly improved quality than baseline methods. Project page: https://soolab.github.io/MVTokenFlow.

## 1 Introduction

With the development of generative artificial intelligence (GenAI) technologies, 2D or 3D content creation (Rombach et al., 2022) already witnessed a huge improvement in recent years. Automatic creation of 4D content Singer et al. (2023b) is an emerging research topic in recent years, which has wide applications in various fields such as AR/VR, video generation, and robotics. However, due to the scarcity of 4D datasets, automatically creating 4D content is still a challenging task.

Due to the huge success of 2D diffusion models for image or video generation, most works (Singer et al., 2023b; Zhao et al., 2023; Bahmani et al., 2024; Zheng et al., 2024) focus on how to utilize these 2D diffusion models for 4D content creation. These works aim to generate synchronized multiview videos for a 3D object and then reconstruct a dynamic 3D representation such as 3DGS or NeRF from the multiview videos. However, such a pipeline mainly faces the challenge of maintaining spatial consistency, that multiview videos are spatially consistent on the same timestep, and temporal consistency, that each video is consistent across different frames on different timesteps. Early stage works (Singer et al., 2023b; Zhao et al., 2023; Bahmani et al., 2024; Zheng et al., 2024; Ling et al., 2024) attain both consistencies by utilizing both spatial and temporal Score Distillation Sampling (SDS) losses. They utilize SDS to distill an image diffusion model (Liu et al., 2023a; Rombach et al., 2022) to create a 3D representation and then animate the 3D representations by distilling a video diffusion model (Blattmann et al., 2023a). Based on these SDS pipelines, some works (Jiang et al., 2023; Yin et al., 2023; Pan et al., 2024) explicitly generate a video as guidance and design new 4D representations to learn better motion fields. However, 4D contents created by these SDS-based methods suffer from low-quality motions and over-saturated appearances. Some very recent works (Zhang et al., 2024; Liang et al., 2024; Li et al., 2024a; Ren et al., 2024; Jiang et al., 2024) directly train a temporally consistent multiview diffusion model from 4D data and reconstruct the 4D representation from the generated multiview images. Though these methods achieve high-quality 4D

---

*Equal contribution.
†Corresponding author.

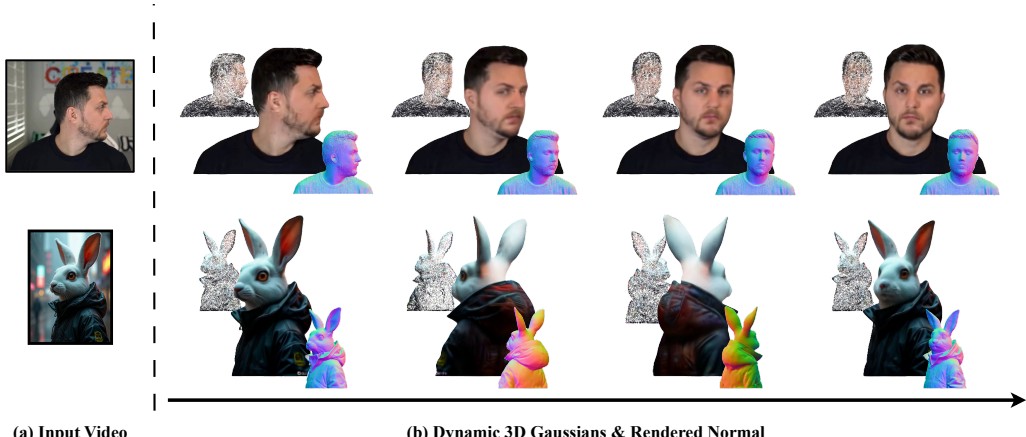

**(a) Input Video**          **(b) Dynamic 3D Gaussians & Rendered Normal**

Figure 1: Given an input monocular video containing a foreground dynamic object (left), MVTo­kenFlow generates a 4D video represented by a dynamic 3D Gaussian field (right) by utilizing a multiview diffusion model and a token propagation method to improve both the spatial and tempo­ral consistency. On the right, we also show the colors of these Gaussian spheres and the rendered normal maps besides the rendered RGB images.

content, they often require a large amount of 4D training data and extensive computation resources for training.

In this paper, we propose a novel 4D generation framework called MVTokenFlow based on mul­tiview diffusion models (Li et al., 2024b) and the token flow technique (Geyer et al., 2023). The challenge in high-quality 4D generation is to create temporally consistent multiview images from monocular 2D videos. Our core motivation is to first construct a coarse 4D field, temporally consis­tent in the front view, and refine it to achieve high temporal consistency across all views. Specifically, in the coarse stage, we utilize multiview diffusion and the temporal consistency of the front view to enhance the spatiotemporally consistent 4D field for better rendered 2D flow. In the refinement stage, this rendered 2D flow guides the regeneration of consistent multiview images, culminating in a high-quality 4D field. As shown in Fig. 1, given an input video that can be generated from a text- or image-to-video generative model, MVTokenFlow generates high-quality 4D contents rep­resented by a dynamic Gaussian field that can be rendered with the splatting technique on arbitrary viewpoints and timesteps. MVTokenFlow achieves spatial consistency by applying the multiview diffusion model to generate multiview images on every frame. The multiview diffusion model is trained on a large 3D dataset to preserve the spatial consistency among all the multiview images. Then, these multiview images will be used in the training of an initial dynamic Gaussian field as the coarse 4D field.

Since the multiview diffusion model is applied to each frame separately, the generated multiview images on different time steps are not temporal consistent with each other, which causes blurry renderings of the coarse 4D field. To improve the temporal consistency between different timesteps, we further apply token flow to associate the generated images of the same viewpoint but on different timesteps. This is based on the fact that a temporally consistent video should share similar tokens among different frames, which has already been utilized by the Token Merging (Li et al., 2024c) or Token Reduction (Geyer et al., 2023) for video editing. To determine the similar tokens on different frames, we utilize the rendered 2D flows from the coarse 4D field to associate pixels from different frames. Then, we regenerate all the multiview images on all timesteps and force the associated pixels to have similar tokens in the reverse diffusion sampling process. This token flow technique effectively improves the temporal consistency of the regenerated multiview images. Finally, the coarse 4D field is refined by the regenerated images to get a high-quality 4D field.

Our contributions can be summarized as follows: (1) We extend the 2D token flow to multiview diffusion to improve temporal consistency. (2) We design a novel pipeline that interleaves dynamic Gaussian field reconstruction with Multiview Token flow to generate multiview consistent videos. (3) We have shown improved quality in 4D content creation from a monocular video. We conduct

experiments on both the Consistent4D (Jiang et al., 2023) dataset and a self-collected dataset to validate the effectiveness of our methods. The results demonstrate that our method generates videos with high-fidelity and high-quality motion on unseen views.

## 2 RELATED WORK

In recent years, diffusion models have gained prominence as a powerful generative framework, excelling in tasks such as image (Rombach et al., 2022; Nichol & Dhariwal, 2021; Nichol et al., 2021; Ramesh et al., 2022) and video synthesis (Blattmann et al., 2023b; An et al., 2023; Ge et al., 2023; Guo et al., 2024; Singer et al., 2022). These models generate data by progressively denoising randomly initialized samples until a coherent structure or scene emerges. Leveraging the flexibility and effectiveness of generative models, they have been adapted to a wide range of tasks Zheng et al. (2023); Tang et al. (2023c); Shi et al. (2024a); Tang et al. (2023b;a), including 3D and 4D content generation. In this section, we will review three parts: diffusion models, 4D scene representations, and 4D generation with diffusion models.

**Diffusion for Generation**    Recently, diffusion models, pre-trained on large-scale datasets (Schuhmann et al., 2022), have made significant strides in generating high-quality and diverse visual content for both 2D image and video tasks  (Rombach et al., 2022; Nichol & Dhariwal, 2021; Blattmann et al., 2023b; An et al., 2023; Huang et al., 2024a). Leveraging aligned vision-language representations Shi et al. (2024b); Dai & Yang (2024); Shi & Yang (2024; 2023b;a; 2022), these models can produce various forms of visual content with impressive diversity and realism conditioned on text or images. To adapt 2D diffusion models for 3D generation, some methods utilize Score Distillation Sampling Loss (Poole et al., 2022; Lin et al., 2023; Chen et al., 2023; Wang et al., 2024) to distill 3D priors and train a neural radiance field (Mildenhall et al., 2020) for 3D asset creation. However, this approach often faces challenges such as slow training speeds and multi-face artifacts (Shi et al., 2023). To address these limitations, another strategy involves fine-tuning pre-trained 2D diffusion models to directly generate multi-view consistent images (Shi et al., 2023; Liu et al., 2023a; Long et al., 2024; Liu et al., 2023b; Li et al., 2024b) from large-scale multi-view datasets (Deitke et al., 2023). These images are then processed through 3D reconstruction algorithms (Wang et al., 2021; Kerbl et al., 2023; Liu et al., 2023c) to produce high-quality 3D assets. Despite these advancements, efficiently leveraging these techniques for 4D generation, ensuring both spatial and temporal coherence, remains a challenging problem.

**4D Scene Representation**    Current 4D scene representations can be broadly categorized into two types based on their underlying 3D scene representation: 1) NeRF-based (Mildenhall et al., 2020) and 2) 3D Gaussian Splatting (3DGS)-based (Kerbl et al., 2023). Both approaches extend static 3D scene representations into the temporal domain by introducing deformable fields or animation-driven training frameworks. NeRF (Neural Radiance Fields) was initially proposed to encode the geometry and appearance of static scenes using implicit models with MLPs. Building upon this, many works have extended static NeRF to handle dynamic scenes, either by modeling a dynamic deformation field on the top of a canonical static scene representation (Pons-Moll et al., 2021; Tretschk et al., 2021; Yuan et al., 2021; Park et al., 2021a; Fang et al., 2022) or by directly learning a time-conditioned radiance field (Li et al., 2022; Gao et al., 2021; Park et al., 2021b; Xian et al., 2021). Despite its success, NeRF-based methods often face limitations in training and inference speed, making them less suitable for real-time applications. Recently, 3D Gaussian Splatting (3DGS) has shown impressive performance due to its efficient training and real-time novel view synthesis capabilities. This method represents static scenes as a set of Gaussian primitives and employs a fast Gaussian differentiable rasterizer with adaptive density control. As an explicit representation, 3DGS also simplifies tasks such as scene editing. 3DGS then has been applied to model dynamic scenes with the similar idea of building a deformation field (Luiten et al., 2023; Wu et al., 2024a; Yang et al., 2024b; Zeng et al., 2024; Wu et al., 2024b). For example, Dynamic 3D Gaussians (Luiten et al., 2023) enable the Gaussians to move and rotate over time under local rigid constraints. This approach efficiently models fine details and temporal changes, making it highly effective for 4D content creation. Together, these representations offer a robust framework for generating realistic and temporally coherent dynamic scenes in 4D space, supporting applications such as animation, scene reconstruction, and motion capture.

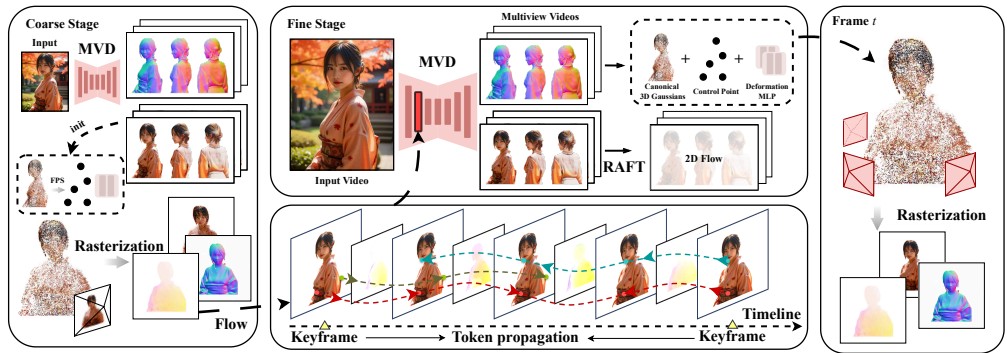

Figure 2: **Overview**. Given an input video that can be generated by video diffusion models, we first apply the Era3D Li et al. (2024b) to generate the multiview-consistent images and normal maps for each timestep. Then, we reconstruct a coarse dynamic 3D Gaussian field field from the generated multiview images. After that, we use the coarse dynamic 3D Gaussian field to render 2D flows to guide the re-generation of the multiview images of Era3D, which greatly improves the temporal consistency and image quality. Finally, the regenerated images are used in the refinement of our dynamic 3D Gaussian field to improve the quality.

**4D Generation** By efficiently integrating advanced diffusion techniques with 4D scene representations, significant progress has been made toward 4D generation. One approach in this direction leverages Score Distillation Sampling (Poole et al., 2022) to distill spatial and temporal prior knowledge from multiple diffusion models into a 4D scene representation, producing spatially and temporally consistent 4D objects, including text-to-video and text-to-image generation. A pioneering work, MAV4D (Singer et al., 2023a) introduced a multi-stage training pipeline for dynamic scene generation, utilizing a Text-to-Image (T2I) model to initialize static scenes and a Text-to-Video (T2V) (Singer et al., 2022) model to handle motion dynamics. Building on this paradigm, several methods have sought to improve 4D generation quality by incorporating image conditions (Zhao et al., 2023), hybrid Score Distillation Sampling (Bahmani et al., 2024), strategies that decouple static elements from dynamic ones (Zheng et al., 2024), and related techniques. However, these methods are largely based on NeRF variants, which suffer from issues like over-saturated appearance and long optimization times. To overcome these limitations, Align-Your-Gaussians (Ling et al., 2024) proposed using dynamic 3D Gaussian Splatting (3DGS) (Kerbl et al., 2023) as the underlying 4D scene representation to learn a deformation field (Park et al., 2021a; Pons-Moll et al., 2021), offering faster training and better real-time capabilities. Despite this, the reliance on SDS loss in these methods leads to slow optimization speeds, limiting their applicability in downstream tasks. Another approach uses video as guidance. Several video-to-4D frameworks (Jiang et al., 2023; Yin et al., 2023; Pan et al., 2024) have been introduced that use video inputs as references to guide 4D generation. These methods attempt to generate dynamic scenes by leveraging video-driven information for more precise motion dynamics. Additionally, to ensure multi-view consistency, recent works have focused on retraining multi-view video diffusion models (Zhang et al., 2024; Liang et al., 2024; Li et al., 2024a; Ren et al., 2024; Jiang et al., 2024) with 4D datasets, integrating both spatial and temporal modules. However, these models often require large amounts of data and are computationally intensive.

## 3 METHOD

Given a monocular video with a dynamic foreground object, our target is to reconstruct the dynamic 3D field represented by a static 3D Gaussian field (3DGS) (Kerbl et al., 2023) and a time-dependent deformation field to deform the 3D Gaussian field to a specific timestep. Note that the video can be either provided or generated from a text description or an image by video diffusion models. Fig. 2 illustrates the outline of our pipeline with 3 steps. First, we run a pretrained multiview diffusion model Era3D (Li et al., 2024b) to generate multiview videos on predefined viewpoints (Sec. 3.1). Then, we reconstruct a coarse dynamic 3D field from the generated multiview videos in Sec. 3.2, where we introduce the representation and training process. Finally, we regenerate the multiview

video by combining the pretrained multiview diffusion model with the token flow with the rendered 2D flow from our coarse dynamic 3D field (Sec. 3.3). The regenerated images are used in the refinement of our coarse 4D field to an improved 4D field with better quality and consistency.

## 3.1 Multiview Video Generation

In this stage, we introduce the pretrained multiview diffusion model Era3D (Li et al., 2024b) to generate multi-view videos from the input monocular video, which will be used in the supervision of dynamic 3D Gaussian reconstruction.

**Multiview diffusion model**. Given a reference video, we split it into $N$ frames $I_{\text{ref}}^{(n)} \in \mathcal{R}^{H \times W \times 3}$ with $n = 1, ..., T$. For every frame, we apply the multiview diffusion model Era3D (Li et al., 2024b) to generate multiple novel view images $I^{(n,k)}$ where $k = 1, ..., K$ is an index of the viewpoint. Note that combining all the generated frames on the same viewpoint but different time steps leads to $K$ videos on different viewpoints. Era3D not only estimates the RGB images but also predicts normal maps on these viewpoints. We use the same symbols to denote both the normal maps and RGB images.

**Discussion about temporal inconsistency**. Simply applying the multiview diffusion model to every frame maintains the consistency between all images of different viewpoints $I^{(n,1:K)} := \{I^{(n,k)}|k = 1, ..., K\}$ but loses coherence between images $I^{(n_0,k)}$ and $I^{(n_1,k)}$ from two arbitrary different timesteps $n_0$ and $n_1$. The reason is that the multiview diffusion model can only maintain the consistency between different viewpoints but the independent generation on different timesteps leads to temporal inconsistency. Thus, the key problem is to improve the temporal consistency of the generated videos. An important observation is that for two frames of a plausible video, their diffusion features or so-called tokens share a strong similarity and are correlated by the 2D flow between them (Geyer et al., 2023). This attribute has been observed by many video editing papers (Geyer et al., 2023; Li et al., 2024c) to design token merging (Li et al., 2024c) and token flow (Geyer et al., 2023). Based on this observation, we improve the temporal consistency by the following two strategies, i.e. enlarged self-attention layers and token propagation with 2D flows.

**Enlarged self-attention**. As observed by many previous works, enlarging the self-attention of stable diffusion (Rombach et al., 2022) (SD) to all the images of different timesteps is helpful in improving the temporal consistency. The adopted Era3D model is also based on the SD model so we enlarge the self-attention layers to include all timesteps to improve temporal consistency. Specifically, in each self-attention layer of the image $I^{(n,k)}$, we keep the query features unchanged but adopt all the features from images $I^{(m,k)}$ of the same $k$-th viewpoint but different timesteps $m$ as the keys and values. This enlarged self-attention provides free temporal consistency without retraining the diffusion model.

**Token propagation with 2D flows**. To further improve the consistency, for a specific video on a specific viewpoint, we only conduct denoising on several keyframes and then propagate the features (tokens) of keyframes to the rest frames. Although minor inconsistencies may exist in the keyframes, we can still reconstruct a high-quality 4D field because keyframes are derived from a temporally consistent input video, and the dynamic 3D Gaussian field is supervised by the video, smoothing residual inconsistencies. Specifically, to conduct denoising on the video $I_t^{(1:N,k)}$ corresponding to the $k$-th viewpoint to get $I_{t-1}^{(1:N,k)}$, we first sample $M$ equidistant keyframes $\{I_t^{(n_m,k)}|m = 1, 2, ..., M\}$ and we conduct the normal denoising process on all these keyframes to get their denosied $\{I_{t-1}^{(n_m,k)}\}$. We obtain the self-attention features $F_t^{(n_m,k)}$ of all these keyframes. Then, for the rest frames, we propagate the features of keyframes to denoise them. Specifically, for a specific frame $I_t^{(n,k)}$ with $n_{m-1} \leq n \leq n_m$, we utilize the 2D flows $\pi(n_{m-1} \to n)$ and $\pi(n \to n_m)$ to warp the features, resulting warped features $F_t^{(n_{m-1} \to n,k)}$ and $F_t^{(n_m \to n,k)}$. We compute the features on the $I_t^{(n,k)}$ by

$$F_t^{(n,k)} = (1 - \lambda_n) \cdot F_t^{(n_{m-1} \to n,k)} + \lambda_n F_t^{(n_m \to n,k)}, \tag{1}$$

where $\lambda_n = (n_m - n)/(n_m - n_{m-1})$ is a position-dependent weighting parameter. Then, we use these propagated features to denoise these intermediate frames between keyframes. Due to the presence of regions in the video that cannot be covered by optical flow, we only propagate features in the early stages $t \leq \tau$, allowing the diffusion process to add more details and occluded regions.

This token propagation scheme effectively utilizes the redundancy in a video and has the potential to improve the temporal consistency of the generated video. Implementing the token propagation requires the estimation of the 2D flow, which is introduced in the following.

**Estimation of 2D flows**. In the beginning, we only have access to the input video but do not have any information on other unseen viewpoints. Thus, we estimate the 2D flow of the input reference video by RAFT (Teed & Deng, 2020). Then, we use this estimated 2D flow to propagate the features of the video on the first viewpoint, i.e. the front view of Era3D, while for other viewpoints, we apply the full denoising process for all diffusion timesteps. Though only one 2D flow is utilized on the front view, the Era3D will utilize cross-viewpoint attention layers to propagate the consistency of the front view to other views and thus improve the temporal consistency.

## 3.2 RECONSTRUCTION WITH GAUSSIAN SPLATTING

Given the generated multiview videos, in this stage, we aim to reconstruct a dynamic 3D Gaussian field. Our method employs a keypoint-controlled dynamic 3D Gaussian representation (Huang et al., 2024b), comprising a static 3D Gaussian and a time-dependent deformable field. For the static 3D field, we represent the field as a set of 3D Gaussians as proposed in 3DGS (Kerbl et al., 2023). For the deformation field, we adopt the representation from SC-GS (Huang et al., 2024b), which first generates a set of 3D control points by clustering 3D Gaussians, then applies an MLP network to translate and rotate these control points, and finally deforms the 3D Gaussians with these control points. On each control point, we associate a set of learnable radius parameters of a radial-basis-function (RBF) kernel that controls how the impact of the control point on a Gaussian will decrease as their distances increase. We train this dynamic Gaussian field using the generated multiview videos. Besides the rendering loss, mask loss, structural dissimilarity (D-SSIM) (Kerbl et al., 2023) loss, and as-rigid-as-possible (ARAP) loss, we also adopt a normal map loss and a 2D flow loss.

**Flow loss**. The flow loss here is to minimize the difference between the rendered 2D flows and the estimated 2D flows on the front view by RAFT (Teed & Deng, 2020). Specifically, for two timesteps, we project the 3D offset of each 3D Gaussian onto to image plane to get the 2D offset of the 3D Gaussian. Then, we combine these 2D offsets with the same alpha blending method as used in splatting to render a 2D flow map. We minimize the difference between the rendered 2D flow map and the estimated 2D flow map with an L1 loss and skip the invisible regions caused by occlusions.

**Normal loss**. Since the Era3D model also generates normal maps for every viewpoint, we also supervise the dynamic 3D Gaussian field with these normal maps. However, computing normal maps from 3DGS is ambiguous without a clear definition of the normal directions. Instead, we first render the depth maps by alpha blending the depth values of all 3D Gaussians and then compute a normal map from the rendered depth map. Finally, we minimize the difference between the generated normal maps and the rendered normal maps. These normal maps pose a geometric constraint on the dynamic 3D Gaussian field and improve the rendering quality.

## 3.3 REGENERATION WITH 2D FLOWS

In this section, we regenerate the multiview images by Era3D with the help of the coarse dynamic 3D Gaussian field trained in the previous section. Then, these regenerated images are used in the refinement of the coarse dynamic 3D Gaussian field. The coarse dynamic 3D Gaussian field often produces blurry results because the images generated in Sec. 3.1 are not temporally consistent enough. However, we observe that these coarse 3D Gaussian fields already produce a reasonable 3D flow field that could improve the temporal consistency for the multiview generation. Thus, we regenerate all the multiview videos with the help of the 3D flow field and refine the dynamic 3D Gaussian field with the regenerations.

**Regeneration and refinement**. In Sec. 3.1, we only have access to the 2D flow of the front view to guide the generation of the multiview videos so all these unseen views are not well constrained with temporal consistency. With the coarse dynamic 3D Gaussian field, we render 2D flow maps for all viewpoints and then incorporate the token propagation in the generation process as introduced in Eq. (1). By propagating tokens on all viewpoints, we greatly improve the temporal consistency

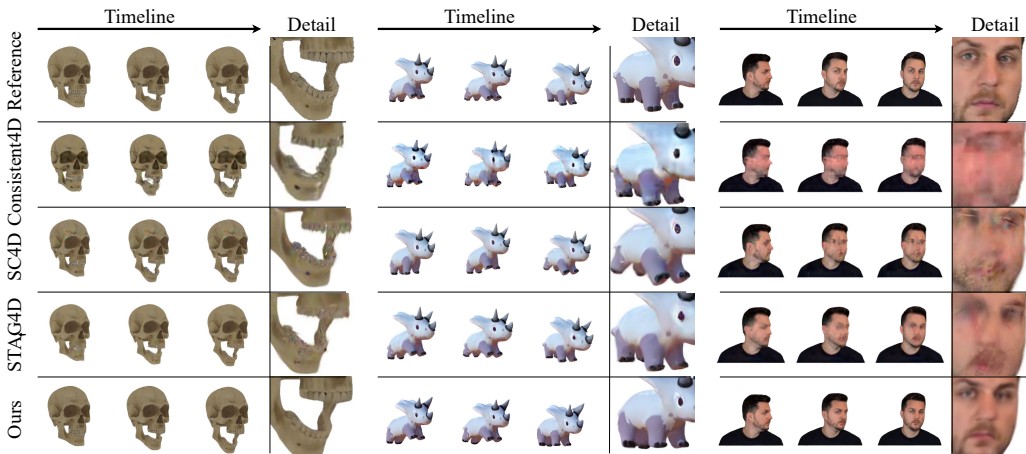

Figure 3: Qualitative comparison on temporal consistency of our method with baseline methods, Consistent4D (Jiang et al., 2023), SC4D (Wu et al., 2024b), and STAG4D (Zeng et al., 2024).

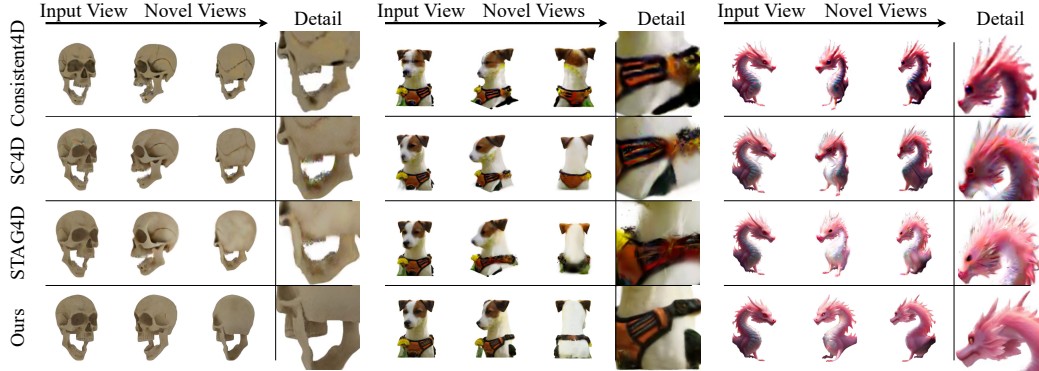

Figure 4: Qualitative comparison on spatial consistency of our method with baseline methods, Consistent4D (Jiang et al., 2023), SC4D (Wu et al., 2024b), and STAG4D (Zeng et al., 2024).

of all generated videos. Then, we utilize the regenerated videos to refine our dynamic 3D Gaussian field, which achieves better rendering quality with less blurry results.

# 4 EXPERIMENT

## 4.1 EXPERIMENTAL SETTINGS

**Implementation details.** For multi-view video generation, we utilize Era3D (Li et al., 2024b) to generate $K = 6$ viewpoints at a resolution of 512x512 for one frame, using 40 denoising steps. We set $\tau = 20$, executing token propagation during the denoising process when $t < \tau$. For keyframe selection, we employ a keyframe interval of 8 frames. In the initialization phase of dynamic 3D Gaussian representation, we initialize 512 control points with Farthest Point Sampling (FPS) sampling. Each Gaussian point is influenced by its 3 nearest control points. During the training of the dynamic Gaussian field, we use an initial learning rate of $3 \times 10^{-4}$ for the MLP, followed by exponential decay. In the refinement phase of the dynamic 3D Gassuian field with regenerated multiview videos, we reset the learning rate to its initial value and applied the same decay strategy. All experiments are conducted on an NVIDIA A40 GPU. We provide more implementation details in the appendix about the losses and data preparation.

**Dataset.** For quantitative comparisons and part of qualitative comparisons, we evaluate our method with the dataset from Consistent4D (Jiang et al., 2023). The dataset comprises 12 synthetic videos

| Method | View Synthesis | | | Spatial Consisency |
|--------|------|------|------|------|
| | LPIPS ↓ | PSNR ↑ | SSIM ↑ | CLIP ↑ |
| Consistent4D (Jiang et al., 2023) | 0.09 | 23.97 | 0.91 | 0.89 |
| SC4D (Wu et al., 2024b) | 0.08 | 29.50 | 0.95 | 0.90 |
| STAG4D (Zeng et al., 2024) | 0.06 | 30.79 | 0.94 | 0.91 |
| Ours | **0.04** | **31.27** | **0.97** | **0.91** |

Table 1: Quantitative results across the three evaluation perspectives, view synthesis, spatial consistency and temporal consistency.

and 12 in-the-wild videos. All videos are monocular, captured by a stationary camera, consisting of 32 frames and lasting approximately 2 seconds. We also collect some videos on the Internet for evaluation.

**Evaluation metrics.** We evaluate our method from three perspectives: consistency with reference videos, spatial consistency, and temporal consistency. Following Consistent4D (Jiang et al., 2023), we utilize PSNR, SSIM, and LPIPS to assess the consistency with reference videos. For multi-view consistency, we employ the CLIP score to measure the semantic similarity of images from different viewpoints.

## 4.2 COMPARISONS

We compare our MVTokenFlow with recent available open-source methods, including Consistent4D Jiang et al. (2023), SC4D Wu et al. (2024b) and STAG4D Zeng et al. (2024). Experiments are conducted on the Consistent4D dataset and a selection of collected videos. This section analyzes representative qualitative results and quantitative results, with additional visualization available in the supplementary videos. We also provide more results of our method in Sec. A.3 of appendix.

**Qualitative Comparison on Temporal Consistency.** Fig. 3 shows the comparison from the perspective of temporal consistency on three samples (*skull, triceratops, and man turning his head*). Consistent4D Jiang et al. (2023) is limited by the expressiveness of cascaded DyNeRF Li et al. (2022), making it challenging to represent the textures of dynamic objects. As a result, the generated outputs are often blurry and distorted, accompanied by artifacts and color discrepancies. SC4D struggles to model motion; as demonstrated in the example of the *triceratops*, its results for the second and third frames are inconsistent with the reference viewpoint due to the weak temporal consistency by simply distilling video diffusion models. For complex real-world scenarios, such as the example of *man turning his head*, early methods are unable to generate coherent and realistic motions. Though recent work (STAG4D) achieves satisfactory temporal consistency in simple samples, it suffers from significant blurriness and distortion when modeling the man turning his head. This failure may be attributed to their reliance on the temporal consistency provided by the single-view reference video, which hinders their ability to learn the correct 3D temporal variations.

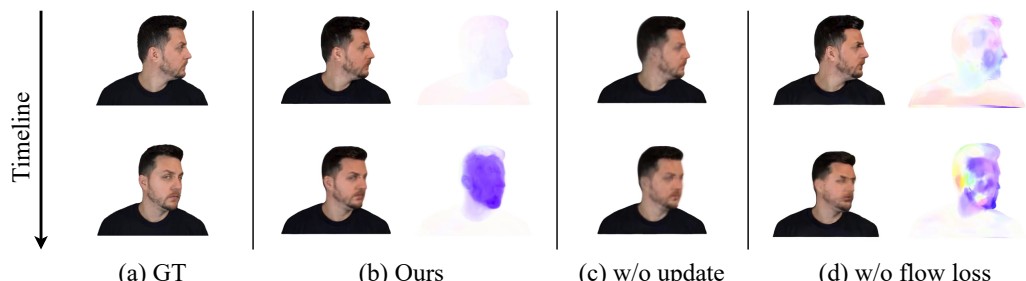

(a) GT      (b) Ours      (c) w/o update      (d) w/o flow loss

Figure 5: Ablation study of the overall architecture. The four parts illustrate (a) Input viewpoint. (b) Our final results. (c) The intermediate outcome from our coarse dynamic 3D field. (d) Result without flow loss.

|              | LPIPS ↓    | CLIP ↑    | FVD ↓      |
|--------------|------------|-----------|------------|
| DG4D         | 0.1748     | 0.915     | 856.86     |
| Consistent4D | 0.1729     | 0.865     | 1072.94    |
| SC4D         | 0.1659     | 0.915     | 879.66     |
| STAG4D       | 0.1506     | 0.885     | 972.73     |
| Ours         | **0.1216** | **0.948** | **846.32** |

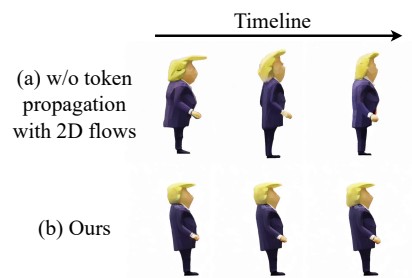

Timeline

(a) w/o token propagation with 2D flows

(b) Ours

Table 2: Quantitative results of novel view synthesis on Consistent4D synthetic objects with multiview videos.

Figure 6: Ablation study on token propagation for multiview video generation.

Compared to previous methods, we adopt flow loss in the generation process, decoupling the learning of motion and appearance. Such a strategy allows our MVTokenFlow to capture high-quality motions while simultaneously modeling clear and detailed appearances.

**Qualitative Comparison on Spatial Consistency.** We also perform comparisons with other methods across different viewpoints to demonstrate spatial consistency of our method, as shown in Fig. 4 with three samples (*skull, dog and dragon*). For novel viewpoints generation, both SC4D and STAG4D tend to produce artifacts, resulting in a significant performance drop compared to the input view. In the example of the *dog*, other methods struggle to maintain the pattern of the bag the dog is wearing, often generating blurred or noisy results. This phenomenon becomes even more pronounced with complex inputs (*dragon*), where previous methods fail to generate eyes that correspond with the input view and similar whiskers. Meanwhile, the generations show over-saturated colors and some artifacts with inconsistent colors.

In comparison, our MVTokenFlow outperforms these methods in both fine-grained spatial consistency and generation quality. These advantages result from our utilization of multiview diffusion combined with normal loss and rendered 2D flow maps for all viewpoints, which leads to strong geometry constraints, whereas previous methods rely on SDS loss as the source of multi-view consistency.

**Quantitative Comparison.** For quantitative comparisons, we evaluate metrics introduced above on Consistent4D dataset and the results are shown in Table 1. Across all metrics, our method consistently outperforms previous approaches including Consistent4D, SC4D and STAG4D. As mentioned in the metric introduction, these results demonstrate that our method exhibits superior consistency with the reference video and enhanced spatiotemporal coherence. Notably, the improvement in LPIPS, which reveals the perceptual consistency of generated images, is particularly significant. This demonstrates that our method can produce more realistic results, aligning with the detailed and accurate textures showcased in our qualitative analysis. Furthermore, as shown in Table 2, we evaluated our method in terms of novel view video synthesis result on Consistent4D synthetic dataset. Our approach exhibits superior temporal consistency under novel views when compared to recent methods.

## 4.3 ABLATION STUDY

To demonstrate the effectiveness of token propagation in the multiview video generation stage and the overall architecture we proposed, we conducted ablation studies for each aspect.

First, we ablate token propagation for multiview video generation as shown in Fig. 6. Without token propagation with 2D flows, both the pose and body shape of the *Trump* exhibit significant variations across different time steps, resulting in a video with flickering changes. In contrast, our proposed method effectively constrains the phenomenon of flickering, significantly improving the temporal consistency of multiview video generation.

Then, we remove the flow loss or skip the regeneration and refinement phase, and compare the results with our full model. To analyze the impact on both motion and appearance, we present both

|  | LPIPS ↓ | FVD ↓ | CLIP ↑ |
|---|---|---|---|
| w/o Enlarged SA | 0.0425 | 348.36 | 0.881 |
| w/o Flow | 0.0408 | 341.50 | 0.887 |
| w/o Noraml | **0.0405** | 320.93 | 0.885 |
| Full | 0.0423 | **313.47** | **0.891** |

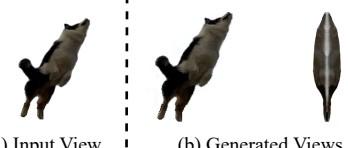

(a) Input View     (b) Generated Views

Table 3: Ablation study on different components in the Consistent4D Dataset.

Figure 7: Limitation on generating novel views for uncommon viewpoints.

rendered RGB images and extracted optical flow at different timesteps in Fig. 5. Comparing (b) with (d), the flow loss significantly enhances the quality of the extracted optical flow, indicating more reliable 3D temporal variations. While (c) displays a relatively blurry image compared to the other experiments, highlighting the significant improvement in generation quality during the regeneration and refinement phase.

Table 3 presents the quantitative results of our ablation study on the Consistent4D Jiang et al. (2023) dataset across three metrics: reference view alignment (LPIPS), temporal consistency (FVD), and multi-view consistency (CLIP). The results demonstrate that flow propagation improves temporal consistency, while the normal loss contributes to enhancing multi-view consistency.

## 5   LIMITATIONS

Though MVTokenFlow succeeds in reconstructing a 4D video from a monocular video in most cases, MVTokenFlow is limited by the ability of the multiview diffusion model, i.e. Era3D (Li et al., 2024b), which may have difficulty in handling complex objects and uncommon viewpoints, as shown in Fig. 7. Improvements in multiview diffusion models could alleviate this problem.

## 6   CONCLUSION

In this paper, we introduce a new pipeline, called MVTokenFlow , to generate 4D videos from just a monocular video. The main challenge in 4D content creation is to simultaneously keep the spatial consistency and the temporal consistency in the generations from diffusion models. Our key idea is to adopt the multiview diffusion models to generate multiview consistent images and then apply the 2D flows to guide the generation of images of different frames to improve temporal consistency. We utilize the information redundancy in a coherent video by adopting the 2D flows to reuse tokens from different frames to generate content for a specific frame. This token re-usage greatly improves the coherence and temporal consistency of the generated images from multiview diffusion models. Experiments demonstrate the effectiveness of our design and show improved quality than baseline methods.

**Acknowledgment:** This work was supported by the National Natural Science Foundation of China(No.62206174) and MoE Key Laboratory of Intelligent Perception and Human-Machine Collaboration (ShanghaiTech University).

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

## A  APPENDIX

### A.1  ADDITIONAL IMPLEMENTATION DETAILS

**Loss functions.** As introduced in Sec. 3.2, we employed several losses during the training of the dynamic Gaussian field, including rendering loss, mask loss, DSSIM loss, ARAP loss, normal map loss, and 2D flow loss. We denote these loss as $L_r$, $L_m$, $L_{DSSIM}$, $L_{arap}$, $L_n$, and $L_f$, respectively. Thus, our loss function can be expressed as $L = \lambda_r L_r + \lambda_m L_m + \lambda_{DSSIM} L_{DSSIM} + \lambda_{arap} L_{arap} + \lambda_n L_n + \lambda_f L_f$, where $\lambda$ represent hyperparameters. For general cases, we set $\lambda_r$ to 0.8, $\lambda_{DSSIM}$ to 0.2, $\lambda_m$ to 2, and the remaining hyperparameters to 1.

**Training iterations.** Following (Kerbl et al., 2023), our training consists of a total of 30K iterations. We first use 5K iterations to learn a static 3D Gaussian from multiview images of a keyframe, which serves as the initialization for the dynamic 3D Gaussian representation. Next, we utilize 10K iterations to learn a coarse dynamic 3D Gaussian field from multiview videos. After regenerating the multiview videos with improved quality, we perform 15K iterations to refine and obtain the final dynamic 3D Gaussian field. The diffusion process requires approximately 30 GB of GPU memory, while the Gaussian field reconstruction utilizes around 10 GB. The comparison of training times with other methods is shown in Table 4.

| | Consistent4D | SC4D | Stag4D | DG4D | Diffusion[2] | Ours |
|---|---|---|---|---|---|---|
| Time | 120 mins | 30 mins | 90 mins | 11 mins | 12 mins | 120 mins |

Table 4: Training time Comparison with other methods. The number of other methods presented in the table is sourced from Yang et al. (2024a).

**Sampling strategy for training the dynamic 3D Gaussian field.** During each training step of the dynamic 3D Gaussian field, we sample various timesteps and utilize all corresponding multiview images for training. Specifically, we employ paired flow inputs to compute the flow loss. However, when the intervals between sampled timesteps are large, rapid movements or changes increase the risk of unreliable flow, which is detrimental to training the dynamic 3D Gaussian field. To mitigate this issue, we implemented an imbalanced sampling strategy that increases the probability of sampling adjacent frames simultaneously.

### A.2  CUSTOM DATA PREPARATION

We manually select high-quality clips from web-collected video data, ensuring each clip contains fewer than 64 frames. Dynamic objects of interest are segmented using SAM2, after which we apply padding to the segmented objects before inputting them into MVD. The resulting videos typically span approximately 4 to 5 seconds.

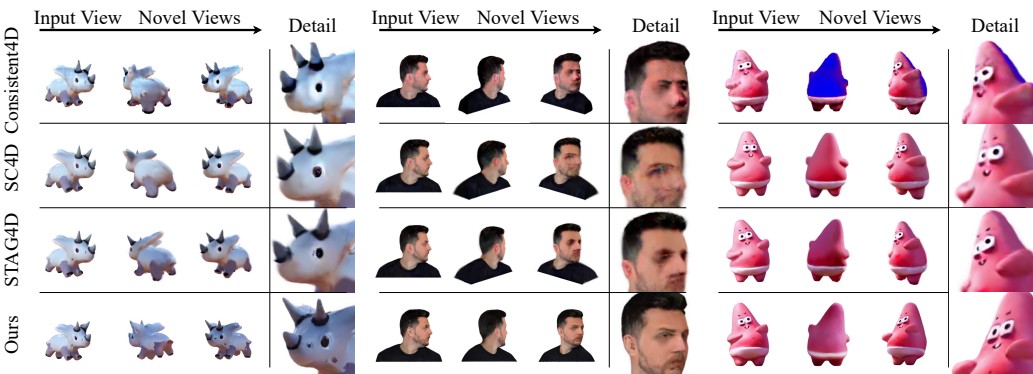

Figure 8: Additional results on spatial consistency with baseline methods, Consistent4D (Jiang et al., 2023), SC4D (Wu et al., 2024b), and STAG4D (Zeng et al., 2024).

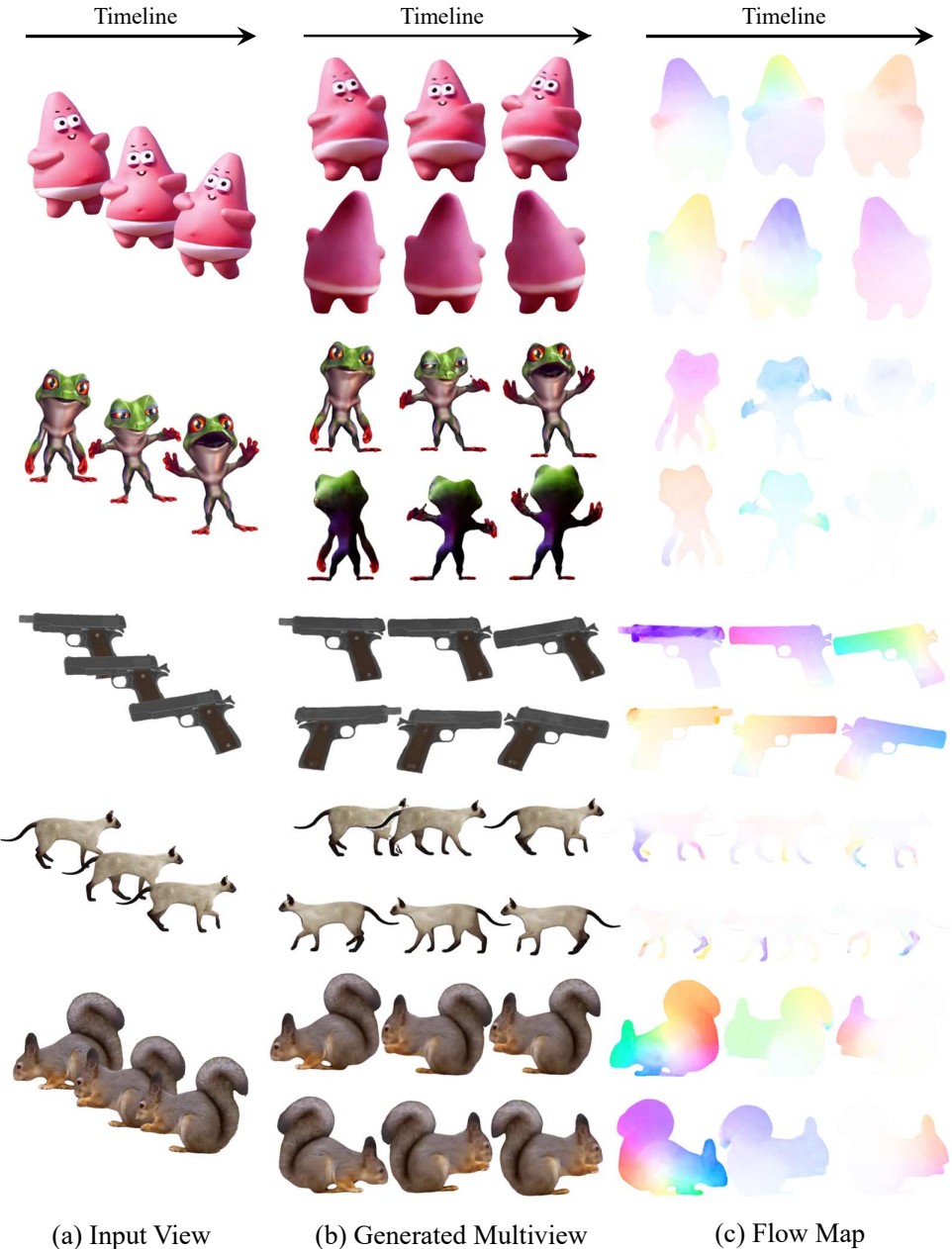

(a) Input View     (b) Generated Multiview     (c) Flow Map

Figure 9: Additional results with flow map of samples in Consistent4D.

## A.3 ADDITIONAL RESULTS

We show additional results on the Consistent4D dataset and a self-collected dataset including the novel-view images and the rendered 2D flow maps in Fig. 9 and Fig. 10. In addition, results with more viewpoints are presented in Fig. 8 and Fig. 11. For the self-collected dataset, we use videos collected from the Internet.

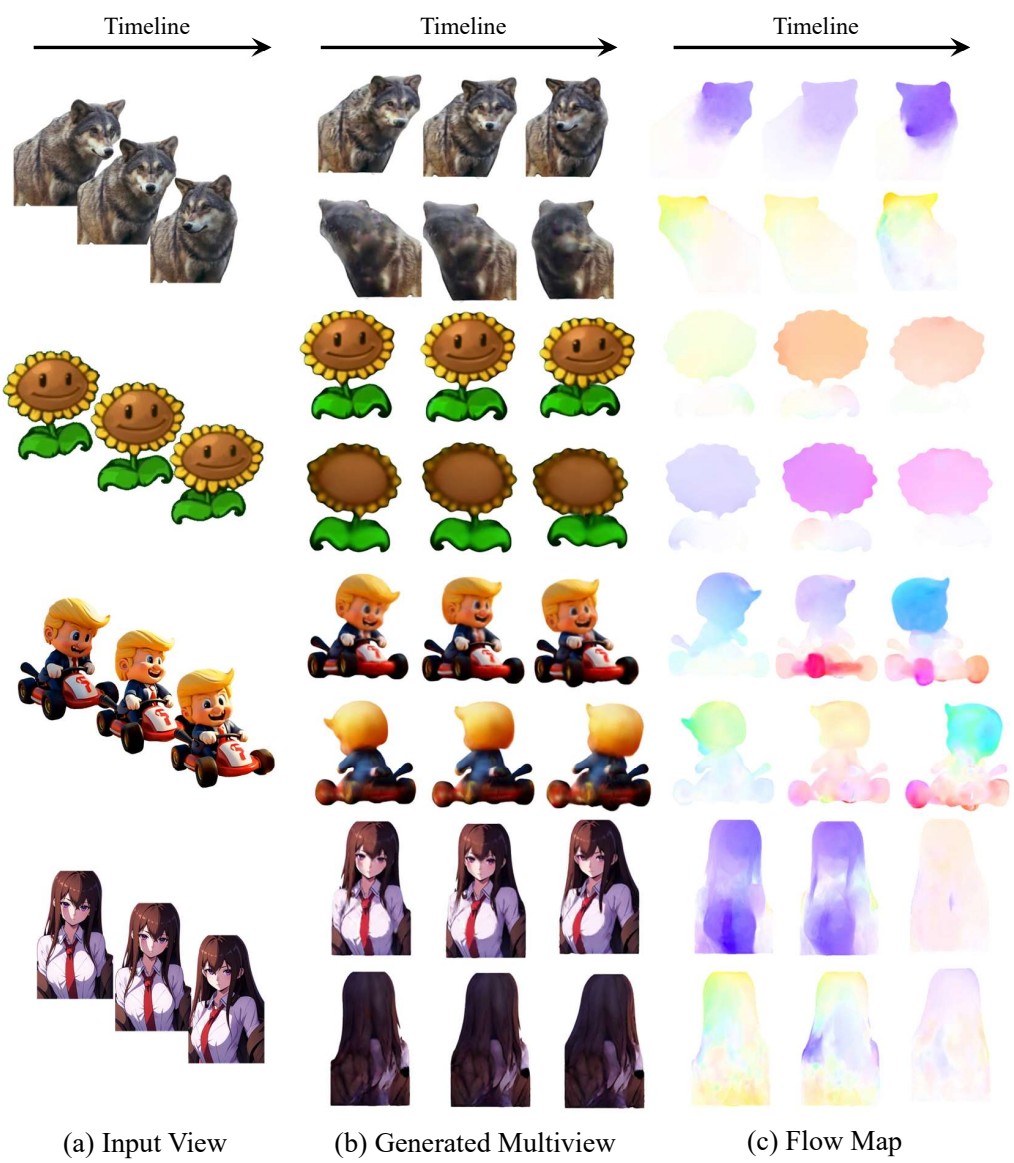

Figure 10: Additional results with flow maps of samples collected on the Internet.

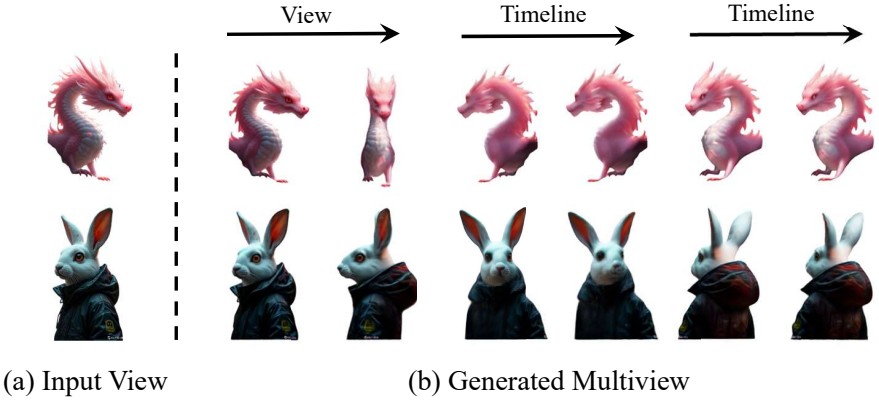

Figure 11: Additional results with more viewpoints of samples collected on the Internet.

