# OpenReview forum: "MVTokenFlow: High-quality 4D Content Generation using Multiview Token Flow"
_ICLR.cc/2025/Conference — ICLR 2025 Poster_

### Official Review · Reviewer_f8rM · 2024-11-01

**Soundness:** 2
**Presentation:** 2
**Contribution:** 2
**Rating:** 6
**Confidence:** 4

**Summary:**

This paper proposed the enlarged self attention on the temporal dimension and incorporate the flow to warp the feature as the additional constraint to improve the temporal consistency. Moreover, the regeneration and refinement have been used to further improve the final performance. The experiment has been on Consistent4D dataset to demonstrate the performance.

**Strengths:**

1. The enlarged self attention on the temporal dimension to maintain the temporal consistency.
2. Using 2D flow to warp the feature as the additional constraint to improve the temporal consistency.
3. The regeneration and refinement have been used to further improve the final performance.
4. The experiment has been on Consistent4D dataset to demonstrate the performance.

**Weaknesses:**

1. The author does not present clearly about the motivation and main contributions of this work in the introduction part.
2. In the related work, the author firstly claimed the diffusion model applied in 4D generation will be discussed, while in the section, 4D Scene Representation, only NeRF and 3DGS based method have been discussed. The related works need to be reorganized to ensure the logical coherence between each parts.
3. Applying the self attention on the temporal dimension has been well studied and proved to be effective way to maintain the temporal consistency. Incorporating this strategy may not have enough novelty.
4. No need to further training not have enough novelty since building the original temporal multi view diffusion model requires huge training while no need for retraining for enlarging the self attention is straight forward.

**Questions:**

Please refer to the Weakness.

---

> ### Author Response · Authors · 2024-11-22
>
> Thank you for the helpful review. We carefully address your questions and comments below.
>
> **Q1**: The author does not present clearly about the motivation and main contributions of this work in the introduction part.
>
> **A1**: The challenge in high-quality 4D generation is to create temporally consistent multiview images from monocular 2D videos. Our core motivation is to first construct a coarse 4D field, temporally consistent in the front view, and refine it to achieve high temporal consistency across all views. Specifically, in the coarse stage, we utilize multiview diffusion and the temporal consistency of the front view to enhance the spatiotemporally consistent 4D field for better rendered 2D flow. In the refinement stage, this rendered 2D flow guides the regeneration of consistent multiview images, culminating in a high-quality 4D field.
>
> Our contributions can be summarized as follows.
>
> 1. Based on our motivation, we design a novel pipeline that interleaves dynamic Gaussian field reconstruction with multiview token flow to generate multiview consistent videos. To the best of our knowledge, we are the first to adopt such an interleaved refinement of 2D flow and multiview token flow for 4D generation.
> 2. We are the first to extend the idea of 2D token flow to multiview diffusion to improve temporal consistency.
> 3. We have demonstrated improved quality in 4D content creation from a monocular video, enhancing both quantitative and qualitative evaluations, and without the need for any bells and whistles or post-process optimization.
>
> **Q2**: In the related work, the author firstly claimed the diffusion model applied in 4D generation will be discussed, while in the section, 4D Scene Representation, only NeRF and 3DGS based method have been discussed. The related works need to be reorganized to ensure the logical coherence between each parts.
>
> **A2**: Thanks for the comment. 4D generation methods are discussed in L150. The related works section discusses 1) diffusion methods, 2) 4D representations, and 3) 4D content generation with diffusion models. The third part is about the diffusion model for 4D generation. We have improved the logical coherence here in the revision.
>
> **Q3**: 1) Applying the self attention on the temporal dimension has been well studied and proved to be effective way to maintain the temporal consistency. Incorporating this strategy may not have enough novelty. 2) No need to further training not have enough novelty since building the original temporal multi view diffusion model requires huge training while no need for retraining for enlarging the self attention is straight forward.
>
> **A3**: Thanks for your comment. **While enlarging self-attention is an important component of our method, it is not the main contribution of our method.** Our key idea is to generalize the token flow method to 4D content creation. By interleaving Gaussian reconstruction and token flow, we enable high-quality 4D content creation. We have included further clarification of our motivation and contributions in the introduction.

---

> > ### Comment · Reviewer_f8rM · 2024-11-24
> >
> > Thanks for your detailed feedback! I appreciate your revision on the paper and I decide to raise my rating to 6.

---

> > > ### Author Response · Authors · 2024-11-25
> > >
> > > We thank the reviewer for the quick reply and consideration! Your detailed feedback is invaluable in helping us refine and further develop our work. We sincerely appreciate your time and effort in reviewing our paper!

---

### Official Review · Reviewer_4Rtf · 2024-11-02

**Soundness:** 3
**Presentation:** 2
**Contribution:** 2
**Rating:** 6
**Confidence:** 5

**Summary:**

The paper addresses the problem of multi-view and temporal consistency in video-to-4D content generation. The proposed method, MVTokenFlow, adopts Era3D for multi view video generation to create coarse 3DGS, and TokenFlow to enhance the temporal consistency of the coarse 3DGS. With this method, the authors are able to generate 4D content while preserving the detail of the content over the timeline. Experiments demonstrate the effectiveness of the proposed method in 4D generation, both quantitatively and qualitatively.

**Strengths:**

(1) The task of video-to-4D content generation, while preserving the details (spatial and temporal information) of an input object, is challenging and well-motivated.

(2) They appropriately chose the multi-view generation method Era3D for spatial consistency and the video generation method TokenFlow for temporal consistency, using the scene representation 3DGS, to construct their pipeline.

(3) The proposed method achieves SOTA performance on video-to-4D generation compared to existing baselines.

(4) The paper is easy to follow.

**Weaknesses:**

(1) Era3D uses Row-wise Multiview Attention (RMA) to achieve multi-view consistency. In L232, the authors say that they use enlarged self-attention to improve the temporal consistency of multi-view video frames. Using $K=6$ viewpoints and a keyframe interval of 8 frames of video, the enlarged self-attention across all different timesteps may be quite memory-heavy. I think this process is a distinguishing feature compared to the previous work Era3D, but the authors don't provide an ablation study on how this component improves temporal consistency in the coarse stage.

(2) As I understand it correctly, the coarse stage produces less temporal-consistent 3DGS, and the fine stage renders 2D flows to guide the re-generation of the multi-view images and create the final temporally consistent 3DGS. When describing this, Fig. 2 does not intuitively illustrate the process. It would be better if the two stages were completely separated.

(3)  The ablation studies on token propagation and flow loss for multi-view video generation show only qualitative results. Quantitative results (using the same metrics as in Tab. 1) are needed to show the generality of these modules.

(4) Similar to point (3), the authors use the normal loss, but this feature is not ablated, either qualitatively or quantitatively.

**Questions:**

(1) How long does it take for the coarse stage and the fine stage, and how much GPU memory is required for each stage? Also, how about the other baselines?

---

> ### Author Response · Authors · 2024-11-22
>
> Thank you for the helpful review. We carefully address your questions and comments below.
>
> **Q1**: Era3D uses Row-wise Multiview Attention (RMA) to achieve multi-view consistency. In L232, the authors say that they use enlarged self-attention to improve the temporal consistency of multi-view video frames. Using viewpoints and a keyframe interval of 8 frames of video, the enlarged self-attention across all different timesteps may be quite memory-heavy. I think this process is a distinguishing feature compared to the previous work Era3D, but the authors don't provide an ablation study on how this component improves temporal consistency in the coarse stage.
>
> **A1**: Thanks for the comment. Our enlarged multiview attention does not bring too much memory consumption. Given the tensor (T,N,H,W) with T frames and N views. The RMA of Era3D is applied on the N dimension to maintain multiview consistency while our enlarged self-attention is applied on the T dimension to associate frames of different timesteps. These are two different attention layers in the implementation, which does not result in too much memory consumption. Our enlarged self-attention layers are orthogonal to RMA of Era3D with different functions. In Table 3, we include an ablation study that removes enlarged self-attention. The FVD results show that enlarged self-attention alone provides limited benefits without our proposed multiview token flow.
>
> **Q2**: As I understand it correctly, the coarse stage produces less temporal-consistent 3DGS, and the fine stage renders 2D flows to guide the re-generation of the multi-view images and create the final temporally consistent 3DGS. When describing this, Fig. 2 does not intuitively illustrate the process. It would be better if the two stages were completely separated.
>
> **A2**: Thanks for the suggestion and we have modified Fig.2 in the revised version.
>
> **Q3**: The ablation studies on token propagation and flow loss for multi-view video generation show only qualitative results. Quantitative results (using the same metrics as in Tab. 1) are needed to show the generality of these modules.
>
> **A3**: Thanks for the suggestion. We have added the quantitative results in Table 3.
>
> **Q4**: Similar to point (3), the authors use the normal loss, but this feature is not ablated, either qualitatively or quantitatively.
>
> **A4**: Thanks for the comment. We have included an ablation study on the normal loss as shown in Table 3 on the Consistent4D dataset. The results show that the normal loss effectively constrain the geometry and thus improve the rendering quality.
>
> **Q5**: How long does it take for the coarse stage and the fine stage, and how much GPU memory is required for each stage? Also, how about the other baselines?
>
> **A5**: The whole process costs about 2 hours. The diffusion process costs about 30G while Gaussian field reconstruction costs about 10G memories. We have included the comparison of generation process time with other methods in Table 4 of the appendix.

---

> > ### Comment · Reviewer_4Rtf · 2024-11-26
> > **Response to rebuttal**
> >
> > Thank you for conducting additional experiments during the rebuttal period. However, after seeing reviewer wwg8's concerns, I find myself in agreement. It would be helpful if the authors could provide further analysis on quantitative evaluation (FVD of other baselines) and their perspective on this matter.

---

> > > ### Author Response · Authors · 2024-11-26
> > >
> > > Thank you for the reviewer's response and further concerns. We have now incorporated the requested results for novel view synthesis on the Consistent4D dataset.
> > >
> > > The results are as follows:
> > > |              | LPIPS↓      | CLIP↑     | FVD ↓   |
> > > | ------------ | ---------- | --------- | ---------- |
> > > | DG4D         | 0.1748     | 0.915     | 856.86     |
> > > | Consistent4D | 0.1729     | 0.865     | 1072.94    |
> > > | SC4D         | 0.1659     | 0.915     | 879.66     |
> > > | STAG4D       | 0.1506     | 0.885     | 972.73     |
> > > | Ours         | **0.1216** | **0.948** | **846.32** |
> > >
> > > These results further demonstrate that our method surpasses previous approaches in terms of visual quality (LPIPS, CLIP) and temporal consistency (FVD). Additionally, Figures 4 and 8 in our paper illustrate that our method achieves higher spatial consistency across different viewpoints and produces more detailed results compared to Consistent4D, SC4D and STAG4D. Furthermore, Figure 3 shows that our method maintains better consistency with the reference video.
> > >
> > > We hope this addresses your concerns and enhances the completeness of our submission. Please let us know if you have any further questions or suggestions.

---

> ### Comment · Reviewer_4Rtf · 2024-11-29
>
> Thank you for sharing the results. Since all my concerns have been resolved, I raise my rating from 5 to 6.

---

> > ### Author Response · Authors · 2024-11-29
> >
> > We thank the reviewer for recognizing our contributions! Your detailed feedback is invaluable in helping us refine and further develop our work. We sincerely appreciate your time and effort in reviewing our paper!

---

### Official Review · Reviewer_wwg8 · 2024-11-02

**Soundness:** 2
**Presentation:** 2
**Contribution:** 2
**Rating:** 6
**Confidence:** 5

**Summary:**

This work focuses on dynamic object generation from a monocular video. The authors propose a two-stage approach, enabling a 3D diffusion model to generate temporally coherent multi-view videos based on reference view video. Specifically, they expand self-attention to all images of different timestamps and generate pseudo multi-view videos in the first stage. The pseudo videos are utilized to reconstruct coarse dynamic gaussians with motion trajectories. In the second stage, they propose a token propagation technique based on 2D optical flow rendered by dynamic gaussians, which helps the 3D diffusion model generate more temporally coherent videos. Experiments on public datasets validate the effectiveness of the proposed framework.

**Strengths:**

1) It is natural to incorporate knowledge of video generation and editing methods into 4D generation. TokenFlow is a reasonable attempt.
2) The authors conduct comparisons and ablation studies to support their claims.

**Weaknesses:**

1. Token Propagation: As illustrated in Section 3.1, token propagation is only conducted between key frames and non-key frames, which means the temporal consistency among key frames is not guaranteed. Although the authors utilize enlarged self-attention for key frames, I think this cannot ensure full temporal consistency, especially when the objects have large movements.

2. Experiments:

a) Dataset: The authors are encouraged to conduct qualitative experiments on the official Consistent4D dataset instead of its demo videos. The official dataset contains 7 synthetic objects with multi-view videos; please refer to the SV4D paper for the performance of other video-to-4D methods on that dataset.

b) Comparison: The authors are encouraged to conduct novel-view synthesis comparisons with other methods if possible.

c) Ablations: The authors are encouraged to conduct quantitative ablations and provide more qualitative analyses.

d) Visualizations: The supplementary videos have only two views of the object, one of which is the input view, which does not provide readers with a full understanding of the performance of the proposed method. The authors are encouraged to provide videos with more views. Additionally, in Figure 3, the comparison with state-of-the-art methods is conducted in the input view, which may not be appropriate.

**Questions:**

The authors claim they compare with all open-sourced methods, but as far as I know, there are other open-sourced video-to-4d works which are not included for comparison, for example, DG4D and Diffusion^2. I'm not asking the authors to comapre with all open-sourced works, instead, I suggest to modify the inappropriate expression.

Besides, I'm willing to raise my score if the authors address my concerns during rebuttal.

---

> ### Author Response · Authors · 2024-11-22
>
> Thank you for the helpful review. We carefully address your questions and comments below.
>
> **Q1**: Token Propagation: As illustrated in Section 3.1, token propagation is only conducted between key frames and non-key frames, which means the temporal consistency among key frames is not guaranteed. Although the authors utilize enlarged self-attention for key frames, I think this cannot ensure full temporal consistency, especially when the objects have large movements.
>
> **A1**: Thanks for your comments. We agree that even after the multiview token flow, there are still some remaining inconsistencies on keyframes. However, we can still reconstruct a high-quality 4D field because
>
> 1. All the keyframes are generated from the input video by multiview diffusion. Since input video is temporally consistent, these keyframes do not show too strong inconsistency from each other.
> 2. We reconstruct a dynamic 3D Gaussian field from these images, which are supervised by the temporal consistent input video and thus can smooth the remaining inconsistency.
>
> We have added this discussion to the paper in L251 of the Method section.
>
> **Q2**: Dataset: The authors are encouraged to conduct qualitative experiments on the official Consistent4D dataset instead of its demo videos. The official dataset contains 7 synthetic objects with multi-view videos; please refer to the SV4D paper for the performance of other video-to-4D methods on that dataset. **Q3**: Comparison: The authors are encouraged to conduct novel-view synthesis comparisons with other methods if possible.
>
> **A2、A3:** Thanks for the suggestion. We have added the quantitative results in Table 2 L442. The results show that our method achieves slightly better performance than Diffusion^2. These results follow the same evaluation setting as Consistent4D, which compares the novel-view rendering quality.
>
>
> **Q4**: Ablations: The authors are encouraged to conduct quantitative ablations and provide more qualitative analyses.
>
> **A4**: Thanks for the suggestion. We have conducted additional quantitative ablation studies on the Consistent4D dataset. The results are shown in Table 3 in L493 in the revised paper. The ablation studies demonstrate the effectiveness of our enlarging self-attention layers, multiview token-flow design, and normal loss.
>
> **Q5**: Visualizations: The supplementary videos have only two views of the object, one of which is the input view, which does not provide readers with a full understanding of the performance of the proposed method. The authors are encouraged to provide videos with more views. Additionally, in Figure 3, the comparison with state-of-the-art methods is conducted in the input view, which may not be appropriate.
>
> **A5**: Thanks for the suggestion. We have added the orbit video in the Supplementary Material. Additional Comparisons with baseline methods on novel views are shown in Figure 8 of the main paper.
>
> **Q6**: The authors claim they compare with all open-sourced methods, but as far as I know, there are other open-sourced video-to-4d works which are not included for comparison, for example, DG4D and Diffusion^2. I'm not asking the authors to comapre with all open-sourced works, instead, I suggest to modify the inappropriate expression.
>
> **A6**: Thanks for the suggestion. We have tried our best to reimplement the baseline methods. We agree that some of these works are still not included due to the rapid development of this field. We have added the comparison with DG4D and Diffusion^2 and changed the sentences in the revision.

---

> ### Comment · Reviewer_wwg8 · 2024-11-25
> **Response to author rebuttal**
>
> Thanks for the efforts of the authors in rebuttal. But my concerns are not adequately addressed.
>
> 1) Quantitative evaluation on Consistent4D dataset: The reported results of other methods in Table 2 differs from results in SV4D and also differs from results in their papers. To be honest, it is hard to believe other method performs so bad in terms of FVD metric. Could the authors explain the reason? Do the authors re-implement other methods by themselves or adopt official implementation? Do the authors use official evaluation code? I have run the official implementation of some other methods before and the results of FVD seems much better than what is reported by the authors.
>
> 2) Conflict between ablation and comparison: `ous` in Table 2 and `full` in Table 3 report very different results. Do the authors use the same evaluation data and the same model?
>
> 3) Visualization results: some visualization results appear flat effect, and some have small motions.

---

> > ### Author Response · Authors · 2024-11-25
> >
> > Thank you for the reviewer's prompt response and further concerns. We hope that our responses below can address these concerns.
> >
> > **Q1**: Quantitative evaluation on Consistent4D dataset: The reported
> > results of other methods in Table 2 differs from results in SV4D and
> > also differs from results in their papers. To be honest, it is hard to
> > believe other method performs so bad in terms of FVD metric. Could the
> > authors explain the reason? Do the authors re-implement other methods by themselves or adopt official implementation? Do the authors use
> > official evaluation code? I have run the official implementation of some other methods before and the results of FVD seems much better than what is reported by the authors.
> >
> > **A1**: The number in Table 2 are borrowed from diffusion^2 https://arxiv.org/pdf/2404.02148. We apologize for directly using their reported numbers without reproducing other methods ourselves due to time constraints. The number of our own method were obtained using the official evaluation code. We are currently reproducing the results of other models using their official codes for reevaluation. Once the results are ready, we will respond promptly.
> >
> > **Q2:** Conflict between ablation and comparison: `ous` in Table 2 and `full` in Table 3 report very different results. Do the authors use the same evaluation data and the same model?
> >
> > **A2**: In Table 2, we present the results of the novel view evaluation for Consistent4D synthetic objects, while in Table 3, we show the results on the reference view on the Consistent4D video dataset. Specifically, following your suggestion, in Table 2, we evaluate the performance of the novel view based on the SV4D setup. In Table 3, we follow the SC4D setting and conduct ablation tests on our model.
> >
> > **Q3**: Visualization results: some visualization results appear flat effect, and some have small motions.
> >
> > **A3:** Our method has achieved significant improvements in both visual details and temporal consistency compared to other methods. For instance, Figures 4 and 8 illustrate enhanced detail quality achieved by our method, while Figure 3 highlights its superior temporal consistency. Large motion remains a challenging issue in this field, but what we want to highlight that we introduce a novel pipeline, offering a new perspective for solving 4D generation. We hope this can provide a new direction for future research.

---

> > > ### Comment · Reviewer_wwg8 · 2024-11-25
> > >
> > > if possible, please provide quantitative ablation results on Consistent4D dataset to keep it consistent with the comparison results.

---

> > > > ### Comment · Reviewer_wwg8 · 2024-11-25
> > > >
> > > > By the way, I don't know what is wrong with the results in Diffusion^2. According to my experience, only very bad results will have FVD over 1500 and more. I believe the performance of all methods in the table are better than that.
> > > > Personally, I guess something is wrong when the authors of Diffusion^2 conduct their evaluation.
> > > >
> > > > Please check the reported results in original papers of comparison methods for a reference.

---

> ### Author Response · Authors · 2024-11-26
>
> Thank you for your timely feedback! We have conducted the requested quantitative ablation study on the Consistent4D synthetic objects to align with the comparison results. The results are as follows:
>
>
> |                 | LPIPS  ↓           | CLIP ↑             | FVD  ↓            |
> | --------------- | ----------------- | ----------------- | ---------------- |
> | w/o enlarged SA | 0.1658 | 0.919 | 882.82 |
> | w/o flow        | 0.1390 | 0.944 | 883.27  |
> | w/o normal      | 0.1633 | 0.921 | 848.89 |
> | full            |**0.1216**        | **0.948**            | **846.32**           |
>
> The results in the ablation table indicate that incorporating the normal loss contributes to visual quality of novel views(full vs w/o normal), while the flow improves temporal consistency(full vs w/o flow).
>
> We appreciate your engagement and effort in reviewing our work and look forward to any further suggestions you might have.

---

> > ### Author Response · Authors · 2024-11-26
> >
> > Following up on our earlier reply, we have now included the requested results from the novel view synthesis on Consistent4D dataset.
> > The results are as follows:
> > |              | LPIPS↓      | CLIP↑     | FVD ↓   |
> > | ------------ | ---------- | --------- | ---------- |
> > | DG4D         | 0.1748     | 0.915     | 856.86     |
> > | Consistent4D | 0.1729     | 0.865     | 1072.94    |
> > | SC4D         | 0.1659     | 0.915     | 879.66     |
> > | STAG4D       | 0.1506     | 0.885     | 972.73     |
> > | Ours         | **0.1216** | **0.948** | **846.32** |
> >
> > These results further demonstrate that our method surpasses previous approaches in terms of visual quality (LPIPS, CLIP) and temporal consistency (FVD). Additionally, Figures 4 and 8 in our paper illustrate that our method achieves higher spatial consistency across different viewpoints and produces more detailed results compared to Consistent4D, SC4D and STAG4D. Furthermore, Figure 3 shows that our method maintains better consistency with the reference video.
> >
> > We hope this addresses your concerns and enhances the completeness of our submission. Please let us know if you have any further questions or suggestions.

---

> ### Comment · Reviewer_wwg8 · 2024-11-27
>
> It's glad to see that the proposed method outperform other methods in term of all metrics.
>
> To make this work better, I suggest the authors to perform a detailed analysis of why the proposed method outperform other methods. I guess the significant gain in CLIP and LIPIS is due to the use of advanced 3D diffusion model, Era3D, which enables novel view results with better visual quality and generates normal map for geometric reconstruction. However, previous works all adopt Zero123 as 3D diffusion. (I compare the results in ablation `w/o normal` and comparison `SC4D` to draw this conclusion)
> But the reason might also be that the proposed method uses reconstruction while previous works adopted SDS. Could the authors provide some experiments to explore this?
>
> Additionally, I want to explain why I emphasize FVD metric. FVD evaluates both visual quality and **temporal consistency**, while previously adopted CLIP/LPIPS/SSIM only evaluates visual quality. Temporal consistency is important for 4D generation, especially when you  regard **lifting a 3D diffusion to 4D in training-free way** as your main contribution. To be honest, the gain in FVD in ablation table is not very satisfying. And it's strange that `w/o enlarged SA` lags behind in LPIPS and CLIP, since I think the per-frame 3D diffusion generation results should have better visual quality yet worse temporal consistency. Try to perform a in-depth analysis, please.
>
> I'm willing to raise the rating if the authors conduct in-depth analysis of their model and comparison methods, regardless of whether their conclusions further highlight the contributions of their method.

---

> > ### Author Response · Authors · 2024-11-27
> >
> > Thank you for your prompt response and for taking the time to review our work thoroughly.
> >
> > We are pleased to hear that our quantitative and ablation experiments have addressed your previous concerns. Regarding your new suggestions, we have conducted an in-depth analysis, which is outlined as follows.
> >
> > - The reasons our method outperforms other approaches:
> >     1. Our method is based on multiview diffusion, through which we introduce **multiview flow**, one of our key contributions. Multiview diffusion enables us to impose constraints not only on colors but also on normals, a capability that the Zero123-based methods lack. Furthermore, multiview diffusion serves as the foundation of our multiview flow.
> >     2. Previous methods rely heavily on SDS loss for novel view synthesis. However, relying solely on SDS loss in 4D generation often results in smoother but less detailed outputs. In contrast, our approach integrates **multi-view rendering loss** and **normal loss**, which enhance visual quality and produce finer details.
> >
> > - We strongly agree that temporal consistency is a crucial aspect of 4D generation. This is precisely why we proposed multiview flow to enhance temporal consistency. Regarding your question about `w/o enlarged SA` , our response is as follows:
> >     1. We analyze the drop in LPIPS and CLIP scores `w/o enlarged SA` and find it to be reasonable. The core reason is that our time-dependent deformation field inherently introduces a degree of temporal smoothing. When faced with temporally inconsistent flow supervision, this smoothing effect often results in blurred multiview images, ultimately compromising overall quality. Specifically, we use the multiview images generated by MVD as the supervision to train the dynamic 3D Gaussian. If the multiview videos used for **supervision lack temporal consistency**—despite featuring high-quality individual frames—it becomes **challenging for the dynamic 3D Gaussian to learn effectively**. This results in a decline in both view quality and temporal consistency.
> >     2. With the help of our **multiview flow**, we can simultaneously improve the temporal consistency of both the dynamic 3D Gaussian and the multiview videos, thereby facilitating the training of dynamic 3D Gaussians.
> >
> > We sincerely thank the reviewer once again for thorough and detailed review. Welcome to further discussion.

---

> ### Comment · Reviewer_wwg8 · 2024-11-28
>
> I was expecting to see some additional quantitative experiments to explore what I mentioned before.
> And I think one important experiment is missing, i.e. evaluating the multi-view video from Era3D directly, and conduct ablation on it, given that the main contribution is multi-view video generation not the following 4d reconstruction.
>
> Anyway, I hope you can spend more time to improve this work. I'll raise my rating to encourage the author to further improve the article and bring more insights to the community

---

> > ### Author Response · Authors · 2024-11-29
> >
> > Thank you to the reviewer for the encouragement and for raising the score to 6. We will
> > incorporate the suggested quantitative and qualitative results along with detailed analyses to enhance and refine our revised paper further.
> >
> > Following your suggestion, we conduct an ablation study to evaluate the performance of multi- view videos directly generated by Era3D. The results are as follows:
> > |              | LPIPS↓      | CLIP↑     | FVD ↓   |
> > | ------------ | ---------- | --------- | ---------- |
> > | Era3D(static)         | 0.1803     | 0.9279     | 962.48     |
> > | Era3D + dynamic deformable field | 0.1723     | 0.9173     | 1387.13    |
> > | Ours         | **0.1216** | **0.948** | **846.32** |
> >
> > In the study, "Era3D(static)" represents measurements taken by selecting a frame from Era3D and copying it across multiple timesteps, serving as a baseline for comparison without utilizing the dynamic field. "Era3D+dynamic deformable field" involves generating multi-view videos by directly modeling a dynamic field on multi-view images from Era3D. All approaches, including ours, are evaluated using the Consistent4D dataset.
> >
> > The results underscore that relying solely on multiview diffusion (MVD) to model the dynamic deformation field for 4D generation does not meet expectations and even underperforms compared to "static video". This shortfall is primarily due to the lack of temporal consistency in the multiview images generated directly by Era3D, which hampers effective learning of the dynamic deformation field.
> >
> > However, MVD can serve as a foundation for our proposed **multiview token flow for 4D generation**. When multiview token token flow is integrated in both dynamic Gaussian models and MVD with enlarged SA, the results, especially in terms of temporal consistency, are significantly enhanced.
> >
> > In the revised version, we will also include qualitative examples that visually compare the direct use of MVD with our method, as well as the various ablation models previously added. Thank you for your suggestion to further enhance our paper.

---

### Official Review · Reviewer_FPXC · 2024-11-04

**Soundness:** 3
**Presentation:** 3
**Contribution:** 3
**Rating:** 6
**Confidence:** 4

**Summary:**

This paper presents a method by which multiview images generated by Era3D independently per frame in a video can be adjusted to be temporally consistent.  A coarse dynamic 3DGS reconstruction is made from the initial multiview videos and 2D flows are computed from these videos.  These flow fields are used to ensure tokens associated by flow between frames are similar.  The multiview images are regenerated from these modified tokens and then a final dynamic 3DGS reconstruction is built.

Results are presented on Consistent4D and a self-collected dataset.

**Strengths:**

* method seems relatively clean without unneeded bells and whistles
* qualitative results look good
* quantitative metrics show improvement
* ablation study is presented

**Weaknesses:**

* Minor point, but I'd recommend the authors add a clear list of contributions at the end of the intro.

* Presented videos in the supplemental are pretty limited.  The columns also aren't labeled.  I'm guessing the left most image and flow field correspond to the input and the right most is some arbitrary second view?  It would have been nice to see some sort of orbit around the object as the video plays.

**Questions:**

* Will the custom dataset be made available?

---

> ### Author Response · Authors · 2024-11-22
>
> Thank you for the helpful review. We carefully address your questions and comments below.
>
> **Q1**: Minor point, but I'd recommend the authors add a clear list of contributions at the end of the intro.
>
> **A1**: Thanks for your comments. Our contributions can be summarized as follows.
>
> 1. We extend the 2D token flow to multiview diffusion to improve temporal consistency.
> 2. We design a novel pipeline that interleaves dynamic Gaussian field reconstruction with Multiview Token flow to generate multiview consistent videos.
> 3. We have shown improved quality in 4D content creation from a monocular video.
>
> We have included this part in the revision.
>
>
> **Q2**: Presented videos in the supplemental are pretty limited. The columns also aren't labeled. I'm guessing the left most image and flow field correspond to the input and the right most is some arbitrary second view? It would have been nice to see some sort of orbit around the object as the video plays
>
> **A2**: Thanks for the suggestion. We have added the orbit video in the Supplementary Material.
>
> **Q3**: Will the custom dataset be made available?
>
> **A3**: Yes, we will make all our examples available after accepting.

---

### Meta-Review · Area_Chair_rCBo · 2024-12-19

**Metareview:**

This paper presents a two-stage method for generating 4D content from monocular videos by enabling 3D diffusion models to produce temporally coherent multi-view videos. In the first stage, multiview diffusion models generate multi-view images at different timesteps. In the second stage, a token propagation technique based on 2D optical flow helps the 3D diffusion model generate more temporally coherent videos. The proposed method effectively addresses a challenging task and demonstrates state-of-the-art performance both qualitatively and quantitatively. Concerns regarding the experiments and other issues raised during the review process were effectively addressed in the rebuttal.

**Additional Comments On Reviewer Discussion:**

The main concerns raised by the reviewers are related to the experiments. First, one reviewer suggested adding experiments on the official Consistent4D dataset rather than using its demo videos. The authors addressed this by including the requested experiments in the revision. While there were discussions about whether the experiments were conducted correctly, the reviewer was ultimately satisfied with the results. Another significant issue involved the ablation study, with reviewers recommending a quantitative ablation analysis, which was added in the revision.

One reviewer also expressed concern that token propagation is only performed between key frames and non-key frames. The rebuttal acknowledged this as a potential limitation but explained why the method remains effective. Additionally, as requested by the reviewers, the rebuttal included an analysis explaining why the proposed method outperforms other approaches. There were also several suggestions regarding the presentation, such as adding a list of contributions and clarifying the motivation, which were addressed in the revision.

---

### Decision · Program_Chairs · 2025-01-22

Accept (Poster)